# Evaluation of the Clinical Efficacy of Using Thermal Camera for Cryotherapy in Patients with Total Knee Arthroplasty: A Prospective Study

**DOI:** 10.3390/medicina55100661

**Published:** 2019-09-30

**Authors:** Zekeriya Okan Karaduman, Ozan Turhal, Yalçın Turhan, Zafer Orhan, Mehmet Arican, Mustafa Uslu, Sengul Cangur

**Affiliations:** 1Department of Orthopaedics and Traumatology, Medical Faculty, Duzce University, 81000 Düzce, Turkey; yturhan_2000@yahoo.com (Y.T.); zaferorhan61@yahoo.com (Z.O.); ari_can_mehmet@hotmail.com (M.A.); 2Department of Orthopaedics and Traumatology, Duzce State Hospital, 81000 Duzce, Turkey; dr.ozn@hotmail.com; 3Department of Orthopedics and Traumatology, Isparta City Hospital, 32000 Isparta, Turkey; mustafauslu74@hotmail.com; 4Department of Biostatisitics, Medical Faculty, Duzce University, 81000 Duzce, Turkey; sengulcangur@duzce.edu.tr

**Keywords:** cryotherapy, hemorrhage, pain, thermal camera, total knee arthroplasty

## Abstract

*Background and objectives:* Cryotherapy is a method of treatment using cold application. This study aimed to evaluate postoperative clinical and hematological parameters and pain associated with total knee arthroplasty in patients and compared cryotherapy to the conventional method of cold ice pack compressions. *Materials and Methods:* Between January 2015 and January 2016, 90 patients who underwent total knee arthroplasty for grade 4 gonarthrosis were prospectively evaluated. The patients were divided into three groups (*n* = 30, each): Group 1, cryotherapy was applied in the pre- and postoperative periods; Group 2, cryotherapy was applied only in the postoperative period; and Group 3 (control group), only a cold pack (gel ice) was applied postoperatively. In all groups, pre- and postoperative evaluations at 6, 24, and 48 h, hemorrhage follow-up, knee circumference measurement, visual analog scale pain score, knee circumference, and temperature measured by thermal camera were recorded. *Results:* Of the 90 patients, 10% were men and 90% were women. The mean age was 64.3 ± 8.1 (range: 46–83) years. The patella upper end diameter values were significantly lower in the postoperative period in Groups 1 and 2 than in Group 3 (*p* = 0.003). Hemoglobin levels at 24 and 48 h postoperatively were significantly lower in Group 3 than in Group 1 (*p* < 0.001, each) and Group 2 (*p* = 0.038, *p* < 0.001). At 6, 24, and 48 h follow-ups, pain values were significantly lower in Group 2 than in Group 3 (*p* < 0.001). Preoperative 6, 24, and 48 h temperature values were significantly lower in Group 1 than in Group 3 (*p* < 0.001 for each). It was found that the difference between preoperative and postoperative knee flexion measurements was significantly different in both groups or the difference between the groups was changed in each period (*p* < 0.001). *Conclusions:* Postoperative cryotherapy is a potentially simple, noninvasive option and beneficial for the reduction of reducing pain, bleeding, length of stay, analgesic requirement and swelling after total knee arthroplasty. Moreover, there was no early or late prosthesis infection in cryotherapy groups, which may be considered as an additional measure to prevent prosthesis infection.

## 1. Introduction

Total knee arthroplasty (TKA) provides satisfactory results in patients with advanced stage osteoarthritis who do not respond to conservative treatment. TKA is becoming increasingly common due to increased life expectancy and the advancement of arthroplasty techniques [1].

Edema, blood loss, pain and undesirable clinical conditions are frequently encountered in the early postoperative period after TKA [2,3]. Patients’ quality of life is affected by edema and pain in the early post-TKA period. High levels of pain, blood loss, and edema may have negative results on early knee range of motion (ROM). In the late period, these adverse events have negative effects on implant survival, ROM, and radiological results.

TKA is a major orthopedic surgery that causes tissue damage and inflammation [4,5]. Cold treatment mainly works by increasing the pain threshold and endorphin release. In addition, it reduces edema, inflammation, and muscle spasm due to vasoconstriction, which indirectly reduces pain [6]. On the contrary, cryotherapy pad wrapped completely around the knee allows the treatment effect to cover a larger area than that with a traditional cold pouch. In addition, cryotherapy provides a more efficient cooling effect and maintains the temperature at levels that reduce the formation of edema (Figure 1). Therefore the quality of life of the patient is increasing due to faster mobilization, reduced need for pain killers and reduced length of stay in the hospital. Thus, with the aforementioned varying levels of efficacy, this study aimed to evaluate the effect of cryotherapy on clinical and functional outcomes in patients undergoing TKA and compare cryotherapy to the conventional method of cold ice pack compressions.

## 2. Materials and Methods

### 2.1. Participants

The study design was approved by the Düzce University Clinical Research Ethics Committee (Düzce, Turkey) (No. 2014/97 from 9 December 2014), and the study was performed in accordance with the principles of the Declaration of Helsinki. Informed consent was obtained from the parents/guardians of the patients included in the study.

Between January 2015 and January 2016, 90 patients who underwent TKA for grade IV gonarthrosis were evaluated in a randomized, single-center, prospective study. A randomized controlled trial was conducted on 90 patients who underwent total knee arthroplasty and were distributed blindly and randomly into three equal groups (*n* = 30): Group 1, cryotherapy was applied in the pre- and postoperative periods; Group 2, cryotherapy was applied only in the postoperative period; and Group 3 (control group), only a cold pack (gel ice) was applied postoperatively.

All of the patients included to the study had advanced gonarthrosis (Grade IV). The exclusion criteria were patients who did not agree to participate in the study, patients with inflammatory disease, osteoarthritis < IV, patients undergoing revision TKA, patients with neurological disorders, coagulopathy, deep vein thrombosis, pulmonary embolism risk, untreated diabetes, or untreated hypertension, and patients who took high doses of anticoagulant drugs.

The patients were divided into three main groups. In Group 1, the patients were administered cryotherapy (8–12 °C) around the knee for 4 h before the operation. After the operation, cryotherapy was applied for the first 6 h; on the first postoperative day, it was applied at 2 h intervals. On the second and third postoperative days, it was applied every 6 h for 2 h. In Group 2, preoperative application of cryotherapy was not performed; the postoperative application was the same as that in Group 1. In Group 3 (control group), a cold pack (gel ice) was applied as standard treatment for 20 min every 2 h for 3 days postoperatively. All patients underwent only unilateral primary TKA. Cryotherapy and gel ice application were performed on the skin via anti-embolic socks or bandages. The cryotherapy device (Waegener®, Beerse, Belgium) is composed of a server and c-pad. The c-server drives the c-pad by use of the selected c-protocol. The device enables controlled-temperature modulation with cooling at a specific temperature (11 °C) for a prolonged time (Figure 2). Care was taken to avoid contact with the skin.

### 2.2. Procedures

Cryotherapy application, knee circumference was measured at 6, 24, and 48 h before and after the operation, thermal camera temperature 5 cm proximal to the patella upper pole and 5 cm distal to the patella midline and tibial tubercle. Postoperatively, all patients performed passive range of motion exercises on postoperative day 1. Active movement was applied until the time of discharge, under the supervision of a physiotherapist. Knee circumference measurement, and measurements were performed by a single physician. The cryotherapy device has the capability to adjust the temperature of the fluid that reaches the pad. It is a user-friendly practical device that allows the cooling process to be started, paused, and restarted. The application device is hypoallergenic and allows for wide surface contact around the knee.

Prophylactic antibiotic (cefazolin sodium, 1 g intravenous) was given to all patients before the procedure. The TKA procedure was performed by two experienced surgeons. Based on the biological age of the patients, a ligament protective or ligament cutter design knee replacement was applied before starting the tourniquet surgical procedure. After the evacuation of venous blood with the treated postoperatively by bandaging the operated leg from toes to mid-thigh with elastic bandage over. The bandage was applied using the visual tensioning guide incorporated in each bandage: the bandage was tensioned until the rectangular shapes printed onto the surface assumed a square shape. In accordance with the manufacturer’s instructions, the bandages were applied in order to achieve a pressure of 35 mm Hg. No drain was used. The bandage was maintained for 48 h, and the limb rested on a pillow under the calf between exercise sessions. A long-leg antiembolism stocking was then applied to the limb for the next 4 weeks. In all patients a pneumatic tourniquet was inflated to 360 mmHg and released after dressing.

### 2.3. Outcome Measures

In all groups, hemoglobin (Hb) level was monitored preoperatively and postoperatively at 6, 24, and 48 h. Blood transfusion was not performed unless the patient’s Hb was 7 or below.

The same pain protocol (75 mg diclofenac administered intra-muscularly in every 12 h and 2 g paracetamol administered intravenously in every 4–6 h) was applied to the patients before the visual analog scale (VAS) pain scores were measured. VAS score was measured at 6, 24, and 48 h before and after the operation. Epidural anesthesia was preferred in all groups, and additional analgesic application for pain with a VAS score of 4 was performed with 5 mL of 0.025% bupivacaine via an epidural catheter. Standard postoperative medical treatment was applied to each group.

Knee circumference was measured at 6, 24, and 48 h before and after the operation, 5 cm proximal to the patella upper pole and 5 cm distal to the patella midline and tibial tubercle. Postoperatively, all patients performed passive ROM exercises on postoperative day 1. Active movement was applied until the time of discharge, under the supervision of a physiotherapist.

Temperature was measured immediately after the cryotherapy or ice application at 6, 24, and 48 h before and after the operation with a thermal camera. When measuring with the thermal camera, the mean value was taken from the maximum, minimum, and average temperatures (Figure 3). Patients were monitored for 12–24 months postoperatively.

### 2.4. Statistical Analysis

Statistical evaluations were performed using the SPSS 22 program (Chicago, IL, USA). Values of *p* < 0.05 were considered statistically significant. Descriptive statistics (mean, standard deviation, median, minimum, maximum, interquartile range) of all data in the study were calculated. One-way analysis of variance and the Kruskal-Wallis tests (post hoc: Dunn test) were used for comparisons between groups. If the comparison of the measurement values of the variables between different periods did not provide normality assumption, the generalized estimating equations method was applied, and parameter estimation was obtained by using the most appropriate model (Gamma with log link, post hoc: least significant difference test).

## 3. Results

Of the 90 patients, 10% were men and 90% were women. The mean patient age was 64.3 ± 8.1 years (46–83 years). The sociodemographic and clinical features of the patients are reported in Table 1. In all patients, the hemorrhage drain was taken out 48 h postoperatively. The patients were discharged on the sixth postoperative day. None of the patients experienced any skin complications due to the cryotherapy or gel ice. In Group 1 and 2 patients, there was no need for revision surgery due to infection and mechanical relaxation in the early period. Two-stage revision surgery was performed in one patient in Group 3 due to early infection. No infection was seen in late period prosthesis surgery in all groups.

In this study, there was a significant difference between the groups in terms of length of hospital stay. The duration of stay in group 1 was significantly lower than that of group 2 and group 3 (*p* < 0.001, *p* = 0.011), no significant difference about cost (*p* = 0.314) was found between the groups (Table 1).

Laboratory blood values and comparison results are reported in Table 2. According to the post hoc test results, the Hb values at 24 and 48 h were significantly lower in Group 3 than in Group 1 (*p* < 0.001 for each) and Group 2 as well (*p* = 0.038, *p* < 0.001). The Hb value at 48 h was significantly higher in Group 2 than in Group 3 (*p* < 0.001).

Descriptive values and comparison results of pain and temperature are reported in Table 3. It was found that the difference between the pain measurements measured at preop, 6, 24, 48 h was significantly different in all three groups or the difference between the groups was changed in each period (*p* < 0.001). VAS pain scores measured at 6, 24, and 48 h were significantly lower in Group 1 and Group 2 than in Group 3 (*p* < 0.001). According to the post hoc test, the VAS pain scores measured at 6, 24 and 48 h in Group 1 were significantly lower than those measured in group 3 (*p* < 0.001 for each). In addition, the pain value measured in 6 h in Group 1 was significantly lower than in Group 2 (*p* = 0.003). The pain values measured in Group 2 at 6, 24 and 48 h were significantly lower than those measured in group 3 (*p* < 0.001).

According to the post hoc test results, the temperature values at 6, 24, and 48 h were significantly lower in Group 1 than in Group 3 (*p* < 0.001 for each). In addition, the preoperative temperature values and those at 24 h were significantly lower in Group 1 than in Group 2 (*p* = 0.002, *p* = 0.008).

Descriptive values and comparison results of knee range of motion values are given in Table 4. It was found that the difference between preoperative and postoperative knee flexion measurements was significantly different in both groups or the difference between the groups was changed in each period (*p* < 0.001). Besides the change in knee flexion measured value in group 1 was approximately 117% higher than in group 3 and 114% higher than in group 2 (*p* < 0.001 for each).

Descriptive values and comparison results of patella alignment, patella lower end diameter, and patella upper end diameter values are given in Table 5. According to the post hoc test, the preoperative patella alignment measurement in Group 2 was significantly lower than the value measured in Group 3 (*p* = 0.004). Patella lower end measurement values at 36 and 48 h postoperative in Group 2 were significantly lower than the measured values of Group 1 (*p* = 0.036, *p* = 0.034) and Group 3 (*p* = 0.006, *p* = 0.003). Furthermore, the patella upper end values measured in Group 3 at 24 and 48 h postoperatively were significantly higher than the values measured in Group 1 and Group 2 (*p* < 0.001 for each).

## 4. Discussion

Our study showed that cryotherapy applications in patients undergoing TKA were significantly more effective than gel ice application for the treatment of edema and bleeding in the knee. Furthermore, our results show that cryotherapy is a safe and effective option for the management of patient pain and rehabilitation. The efficacy of cryotherapy treatment for pain and early range of motion was previously reported [7,8]. In our study It was found that the difference between preoperative and postoperative knee flexion measurements was significantly different in both groups or the difference between the groups was changed in each period (*p* < 0.001). Besides the change in knee flexion measured value in group 1 was approximately 117% higher than in group 3 and 114% higher than in group 2.

In published literature, one study reported that pain increased to the highest level within 24 h after TKA [9]. In our study, VAS pain score was found to be higher in group 3 at 24 h postoperative, but lower in group 1 and 2.

Thienpont reported that cryotherapy did not reduce pain and analgesic use [10]. In our study, the need for pain and analgesia in groups 1 and 2 was found to be less in the postoperative period. Showed that cryotherapy reduces pain and reduces the use of analgesia. Pain occurs in response to inflammation. Effective and regular cold administration via cryotherapy inhibits inflammation by reducing cytokine release [2,11,12]. Chughtai et al. suggested that the pain-reducing effects of cryotherapy could reduce the use of addictive painkillers, such as opioids [13]. In our study, VAS pain score was lower in group 1 and 2 compared to control group.

In TKA, perioperative blood loss of 800–1800 ml can occur in spite of tourniquet and gel ice application [14,15]. Blood transfusions increase the risk of periprosthetic infection, as well as cost. In a prospective study including 50 patients reported by Ruffilli, the continuous application of cold therapy and gel ice were compared; no significant difference in blood loss was observed between the groups [16]. Kuyucu et al. also did not find significant differences in blood loss between groups [17]. We had to provide two units of blood supplementation for the patients in Group 3, but we had not provided blood supplementation into other groups. We found that the Hb levels measured in Group 3 were lower than those in Groups 1 and 2 in the postoperative period and that postoperative cryotherapy was effective in reducing blood loss; however, preoperative cryotherapy application had no significant effect.

Some data suggest that cryotherapy and cold application reduces edema around the knee by reducing vasodilation after TKA [18,19]. Moreover, Ruffilli et al. evaluated knee circumference and found no significant postoperative difference in the measurements of 15 upper and lower patella regions [16]. In the present study, significant prevention of swelling, as evidenced by reduced thigh circumference on postoperative day 3, can presumably be attributed to the use of cryotherapy.

In the literature, information about the measurement of temperature differences associated with various cooling methods applied to the knee and the use of thermal cameras is limited. TKA-induced inflammation leads to a significant increase in deep temperature in the knee. Ueyema et al. reported a correlation between temperature increase in the early stage after TKA and knee ROM recovery; this supports the role of cryotherapy for early improvement in ROM after TKA [20]. In the present study, we compared the heat values around the knees among the groups before and after surgery. No significant difference was found in the postoperative period between Groups 1 and 2. However, in Group 3, the temperature values were high. We concluded that maintaining the temperature of cryotherapy around the knees below the body temperature is a more effective treatment for edema, blood loss, and pain than gel ice application.

In secondary osteoarthritis, such as rheumatoid arthritis and traumatic arthritis, TKA due to early degeneration may be a preferred treatment option for younger patients. However, TKA for primary osteoarthritis is primarily performed in older patients. The average age reported for TKA in primary osteoarthritis was 65.14 ± 4.06 (range, 65–69) years [21]. In our study, the mean age was 64.3 ± 8.1 (range, 46–83) years.

Patients with primary knee osteoarthritis have limited physical activity secondary to pain and tend to have higher body mass index (BMI) [22]. In the present study, we also found patients to have a high BMI, with a mean value of 32.9 kg/m^2^ (range, 22.1–57.8 kg/m^2^). Cold applications to the skin can have a limited effect in patients with a high BMI [23]. In cryotherapy applications, we think that this technique can affect a larger tissue area with the addition of a pad that allows constant cold application of the same temperature and is, thus, more effective for overweight patients. The prevelance of TKA in male patients is nearly 10 times lower than the females [24]. Also in our study, the percentage of males was nearly 10% (male: 10.1%, female: 89.9%).

Mumith et al. reported that cryotherapy application had high costs [23]. In the present study, we found no significant additional costs associated with the use of cryotherapy. However, we think that the control of the cryotherapy device during the application requires experienced personnel.

The strengths of the current study include the prospective evaluation of the patients and the measurement of heat exchange by thermal camera. The limitations of this study are the short follow-up period and non-standardization of the knee prosthesis applied to patients. Some further researches are needed in this area to gain sufficient knowledge on other outcomes such as long-term quality of life measures and patient satisfaction.

## 5. Conclusions

Knee replacement is one of the major surgeries of orthopedics; however, there is no consensus on the effectiveness of cryotherapy after knee replacement. In our study, we evaluated the efficacy of cryotherapy with thermal imaging camera and found that keeping the surgical area at the desired temperature was clinically and functionally good choice to other treatment methods. Postoperative cryotherapy is a potentially simple, noninvasive option and beneficial for the reduction of reducing pain, bleeding, length of stay, analgesic requirement and swelling after total knee arthroplasty. Moreover, there was no early or late prosthesis infection in cryotherapy groups, which may be considered as an additional measure to prevent prosthesis infection.

## Figures and Tables

**Figure 1 medicina-55-00661-f001:**
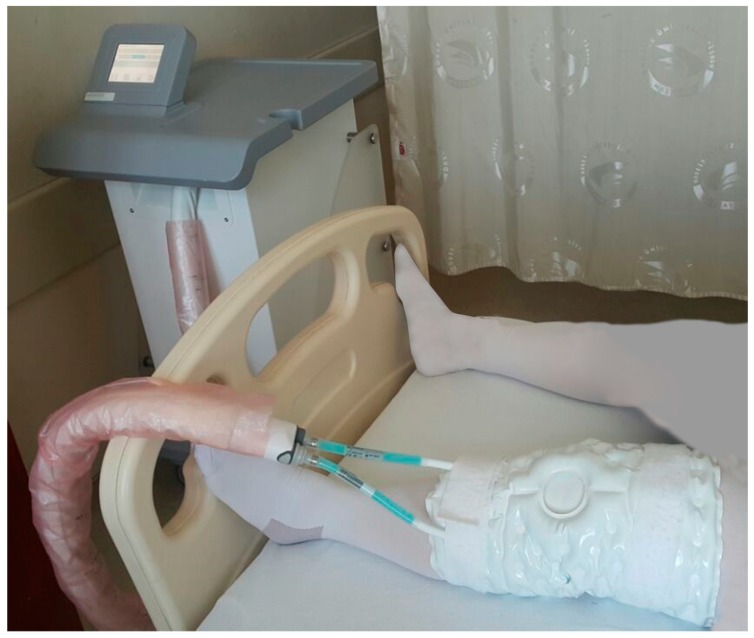
Application of cryotherapy pad on a patient.

**Figure 2 medicina-55-00661-f002:**
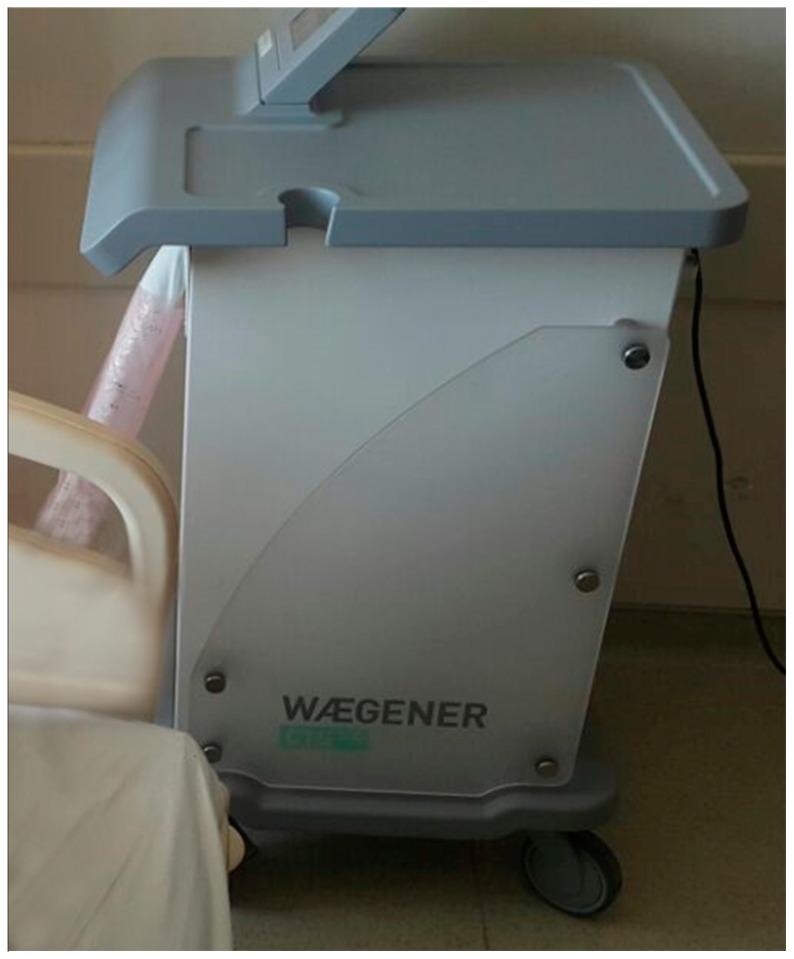
Adjustable cryotherapy device.

**Figure 3 medicina-55-00661-f003:**
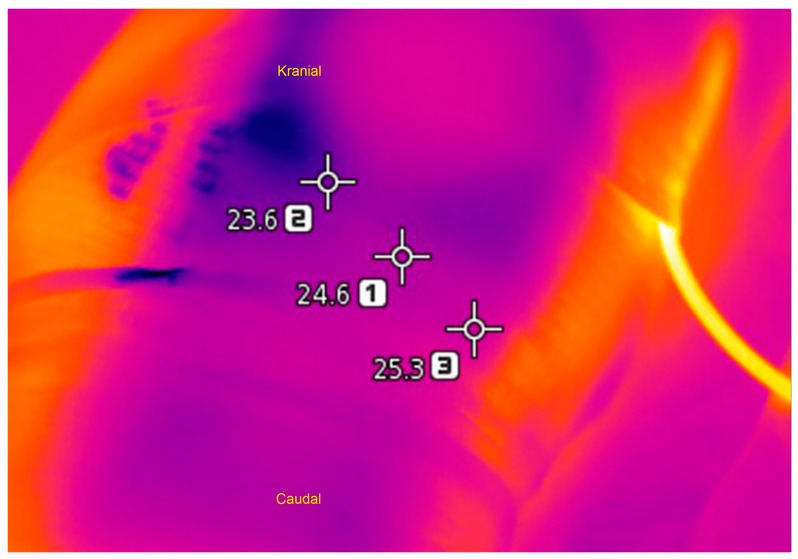
Temperature measurement of knee circumference with thermal camera.

**Table 1 medicina-55-00661-t001:** Clinical and sociodemographic features of patients.

		Group												
		Group 1			Group 2			Group 3			Total			
		*n*	R%	C%	*n*	R%	C%	*n*	R%	C%	*n*	R%	C%	*p*
**Sex**	**Male**	2	22.2	6.7	4	44.4	13.3	3	33.3	10.3	9	100	10.1	0.763
	**Female**	28	35.0	93.3	26	32.5	86.7	26	32.5	89.7	80	100	89.9	
**Side**	**Right**	20	37.0	66.7	19	35.2	63.3	15	27.8	51.7	54	100	60.7	0.469
	**Left**	10	28.6	33.3	11	31.4	36.7	14	40.0	48.3	35	100	39.3	
**Age * (years)**		64.6 ± 6.8 65 (51–79)			62.7 ± 8.8 62.5 (46–80)			65.5 ± 8.6 67 (46–83)			64.3 ± 8.1 65 (46–83)			0.417
**Weight** **^#,&^ (kg)**		81.8 ± 11.9 80 (61–107)			87.1 ± 11.0 85 (73–130)			83.6 ± 8.1 84 (66–98)			84.2 ± 10.6 84 (61–130)			0.159
**Length of patient** **^#^** **(cm)**		1.6 ± 0.1 1.6 (1.5–1.8)			1.6 ± 0.1 1.7 (1.5–1.8)			1.6 ± 0.1 1.6 (1.5–1.9)			1.6 ± 0.1 1.6 (1.5–1.9)			0.204
**BMI ^#,&,£^**		30.8 (25.4–44.1)			31.9 (26.6–57.8)			31.6 (20.4–40)			31.6 (20.4–57.8)			0.485
**Length of stay** **^#^ (days)**		3.4 ± 0.5 3 (3–4)			4.3 ± 0.7 4 (3–5)			6.3 ± 1.0 6 (5–8)			4.6 ± 1.4 4 (3–8)			<0.001
**Cost (TL)** **^#^**		6478.7 ± 1282.4 6266 (4168–12276)			6268.8 ± 251.6 6237 (6035–6851)			6350.6 ± 234.9 6290 (6035–7255)			6366.2 ± 766.8 6271 (4168-12276)			0.314

* Mean ± standard deviation (min–max), ^#^ Median (min-max), R %: Row %, C %: Column %, TL (Turkish lira), BMI (body mass index). ^&^: *p* < 0.01 for Group 1 vs. Group 3, ^£^: *p* < 0.05 for Group 2 vs. Group 3.

**Table 2 medicina-55-00661-t002:** Hemoglobin values according to groups.

	Group	Period	Mean	SD	Median	Min–Max	IQR	*p*
**Hb**	**1**	Preop	12.5	1.3	13.0	9–15	1.3	0.008
		6 h	11.1	1.3	11.0	8–13	2	
		24 h	10.5	1.4	10.0	8–14	2.3	
		48 h	9.8	1.2	10.0	7–12	2	
	**2**	Preop	12.6	1.1	13.0	10–15	1	
		6 h	11.1	1.2	11.0	9–14	2	
		24 h	11.0	1.3	11.0	7–13	2	
		48 h	10.4	1.2	10.0	8–13	1.3	
	**3**	Preop	11.9	1.7	12.0	8–16	2	
		6 h	10.5	1.4	11.0	8–13	2	
		24 h	9.7	1.5	10.0	7–12	2	
		48 h	8.9	1.2	9.0	7–12	1.5	

Hb, hemoglobin; IQR, interquartile range; Max, maximum; Min, minimum; SD, standard deviation.

**Table 3 medicina-55-00661-t003:** Descriptive values and comparison results of pain and temperature measurements.

	Group	Period	Mean	SD	Median	Min–Max	IQR	*p*
**VAS**	**1**	Preop	9.2	0.5	9	8	0.3	<0.001
		6 h	6.1	1.2	6	4		
		24 h	5	1	5	4	1	
		48 h	3.9	0.8	4	2	1.3	
	**2**	Preop	9	0.7	9	8	0.5	
		6 h	6.8	1.2	7	5	2	
		24 h	5.2	0.7	5	4	0.3	
		48 h	4.1	0.7	4	3	0	
	**3**	Preop	8.9	0.8	9	8	1.5	
		6 h	8.9	0.3	9	8	0	
		24 h	7.2	0.7	7	5	1	
		48 h	6.6	0.9	7	4	1	
**Temperature**	**1**	Preop	25.0	3.3	24.0	14–29	5.25	<0.001
		6 h	26.3	2.3	27.0	23–30	4	
		24 h	27.4	2.6	27.5	23–35	4	
		48 h	27.2	2.9	28.0	20–31	3.3	
	**2**	Preop	36.0	1.7	27.0	23–30	2	
		6 h	27.6	1.3	27.5	25–30	2	
		24 h	28.1	1.3	28.0	25–30	2	
		48 h	27.8	1.6	28.0	24–31	2	
	**3**	Preop	35.8	0.4	36.0	35–36	0	
		6 h	35.9	0.5	36.0	35–37	0	
		24 h	35.8	0.9	36.0	34–37	0.5	
		48 h	36.5	0.5	36.0	36–37	1	

IQR, interquartile range; Max, maximum; Min, minimum; SD, standard deviation; VAS, visual analog scale.

**Table 4 medicina-55-00661-t004:** Descriptive values of knee range of motion (ROM) values and comparison results.

	Group	Period	Mean	SD	Median	Min–Max	IQR	OR for Group x Period (95% Wald CI)
**ROM**	**1**	Preop	91.7	8.7	92.5	70–100	10	1.143 ^&^ (1.064–1.228) 1.166 ^$^ (1.084–1.254)
		Postop	109.8	6.1	110	100–120	10	
	**2**	Preop	90.0	14.9	90	60–110	21.3	
		Postop	94.3	11.7	92.5	70–110	21.3	
	**3**	Preop	95.0	4.8	95	85–105	9	
		Postop	97.6	4.2	100	90–105	5	

Group 1: preop-postop cryotherapy, Group 2: Postop cryotherapy, Group 3: Gel SD: Standard Deviation, Min: Minimum, Max: Maximum, IQR: Interquartile Range, OR: Odds Ratio, CI: Confidence Interval, ^&^: Group 1 vs. Group 2, ^$^: Group 1 vs. Group3. X: Group x Period: The interaction effect of Group x Period

**Table 5 medicina-55-00661-t005:** Descriptive values and their comparison between groups.

	Group	Period	Mean	SD	Median	Min–Max	IQR	*p*
**Patella level**	**1**	Preop	41.9	4.4	40.0	37–51	5.8	<0.001
		6 h	43.5	4.5	41.0	39–52	7.3	
		24 h	44.0	4.5	42.5	38–54	7.3	
		48 h	44.2	4.6	42.5	37–55	6	
	**2**	Preop	41.7	2.0	42.0	36–45	3	
		6 h	42.5	1.8	43.0	37–46	3	
		24 h	42.6	2.0	43.0	37–46	3	
		48 h	43.0	2.0	43.0	37–47	4	
	**3**	Preop	43.8	3.3	43.0	39–52	2.5	
		6 h	44.8	3.5	44.0	40–53	4	
		24 h	45.0	3.6	45.0	40–53	4	
		48 h	45.5	3.7	45.0	41–55	4	
**Patella lower end**	**1**	Preop	39.6	4.4	38.0	33–50	3.3	0.275
		6 h	40.5	4.3	39.0	34–50	5	
		24 h	41.1	5.1	39.0	34–55	4.5	
		48 h	41.1	4.3	40.5	33-52	4.5	
	**2**	Preop	38.1	1.9	39.0	30–40	2	
		6 h	39.0	1.5	39.0	36–41	2	
		24 h	39.1	1.3	39.0	37–41	2	
		48 h	39.3	1.3	39.5	37–41	3	
	**3**	Preop	39.3	2.9	39.0	36–48	2	
		6 h	40.4	2.8	40.0	37–48	2.5	
		24 h	40.8	3.0	40.0	37–49	3.5	
		48 h	41.2	3.2	41.0	36–50	2.5	
**Patella upper end**	**1**	Preop	46.1	4.9	44.0	40–57	7.3	0.063
		6 h	47.2	4.8	45.5	41–58	8	
		24 h	47.7	5.6	47.5	39–60	7.3	
		48 h	47.5	5.1	47.0	40–60	7.3	
	**2**	Preop	47.3	2.0	48.0	43–50	3	
		6 h	48.2	2.0	49.0	44–51	3	
		24 h	48.6	2.2	49.0	44–53	3	
		48 h	48.9	2.2	49.0	44-53	2.3	
	**3**	Preop	49.4	3.6	49.0	43–56	4.5	
		6 h	50.4	3.8	50.0	43–57	4.5	
		24 h	50.9	3.7	51.0	44–58	4	
		48 h	51.5	3.9	51.0	44–60	4.5	

SD: standard deviation, Min: minimum, Max: maximum, IQR: interquartile range.

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
