# Peer review of "Evaluation of the Clinical Efficacy of Using Thermal Camera for Cryotherapy in Patients with Total Knee Arthroplasty: A Prospective Study"

_medicina, 2019, doi:10.3390/medicina55100661_

Round 1
Reviewer 1 Report
Dear Colleagues, thank you very much for submitting your paper to Medicina. Cryotherapy seems to be an easy and convenient tool for the early postoperative period with regards to reducing swelling and edema and therefore help in better mobilizing patients as well as increasing their impression of quality of life in the early postoperative period.
Please address the following comments properly:
L 35: TKA already is the standard treatment in OA knees not increasingly common. Please reword.
L 37: delete first “and”
L 40: Not only the ankle ROM but especially the knee ROM is affected as well as the mobilization of the patients. Please include this information. Add “may have” negative results.
L 51: Please outline/sum up (may also be put under “discussion”) the possible influence of increased quality of life, faster mobilization, reduced need of pain killers, probably even reduced length of stay/earlier discharge from hospital – and the need to check this with a larger follow-up as this study is a very short-termed thing.
Figure 1: The pictures seem to be contorted. Please provide better ones.
L 57: Number of the approved design available?
L 59: Also, from the patients themselves?
L 65: add – the standard of care so far
L 66: contradiction with l 60 above – please be more precise: just grade IV or also III
L 66: exclusion criteria would have also been osteoarthritis < III
L 79: Please provide additional information with regards to the cryotherapy device!
Figure 2: Please provide a better picture
L 98: please be more precise – what does include the pain protocol?
Figure 3: Add kranial/caudal etc. to the picture for a better understanding
L 129: Early or late PJI?
Table 1: length of what? Please add the unit. bmi has to be written in capital letters. Correct hospitalization days to LOS (length of stay). Line distance or break after “costs”
Table 2: How does result the overall p of 0.008?
Table 3: See Table 2. Please explain the general p
L 151: Where are the results for the circumferential measurements? Furthermore - are there results for range of motion, at least for the time of discharge? This is a common information also in letters of discharge. Please add information with regards to ROM in the whole manuscript. I would also like to have additional information regarding any difference in the way patients could be mobilized!
L 153: add significantly
L 156: The results allow no real statement with regards to rehabilitation yet.
L 157: Reported where? Be more precise.
L 157: Be more precise with the summarizing aspects of your study.
L 159: Add – as well.
L 160: In Arthroplasty?
L 165: Did you also see an effect with regards to reduced need of pain medication? Please add, if possible
L 166: What is meant with turbiditiy?
L 167: The correct plural is costs; please correct
L 169: In which context? Also arthroplasty? Please me more precise with regards to the overall discussion!
L 194: For all indications of TKA?
L 200: If you discuss the socioeconomic cohort as well, please also add literature information with regards to gender distribution
L 202: costs instead of cost – it usually is used as a plural word
L 203: It may even reduce costs. Discuss this aspect.
L 208: See also above!
L 2010: Reword - … in orthopedic surgery. And add the possible postoperative complications as mentioned in the introduction. Please be more precise in this section with regards to your actual findings.
Discussion in general: Is there no more literature of further studies available? Please be more precise and expand the discussion to a greater academic level.
Author Response
Replies uploaded as files

Reviewer 2 Report
Interesting paper, let down by some lack of clarity over methodology.
It would be important to know whether there was a consistent approach to post operative bandaging. When was the bandage taken off? Was it after 24 or 48hrs, was it before or after the cooling treatment? I presume the cooling treatment was through / over the top of the bandaging? Did every surgeon use single layer or double layer bandaging? i.e. we need much more information here and an idea of consistency.
A note on consistency of tourniquet times between and within groups would be helpful, especially as they inflate their tourniquet up to 360mmHg as this seems much higher than UK practice and in itself will be painful.
Author Response
Replies uploaded as files
